# TOWARDS SKILLED POPULATION CURRICULUM FOR MULTI-AGENT REINFORCEMENT LEARNING

## ABSTRACT

Recent advances in multi-agent reinforcement learning (MARL) allow agents to coordinate their behaviors in complex environments. However, common MARL algorithms still suffer from scalability and sparse reward issues. One promising approach to resolve them is *automatic curriculum learning* (ACL), where *a student* (curriculum learner) train on tasks of increasing difficulty controlled by *a teacher* (curriculum generator). Unfortunately, in spite of its success, ACL's applicability is restricted due to: (1) lack of a general student framework to deal with the varying number of agents across tasks and the sparse reward problem, and (2) the non-stationarity in the teacher's task due to the ever-changing student strategies. As a remedy for ACL, we introduce a novel automatic curriculum learning framework, Skilled Population Curriculum (SPC), adapting curriculum learning to multi-agent coordination. To be specific, we endow *the student* with population-invariant communication and a hierarchical skill set. Thus, the student can learn cooperation and behavior skills from distinct tasks with a varying number of agents. In addition, we model *the teacher* as a contextual bandit conditioned by student policies. As a result, a team of agents can change its size while retaining previously acquired skills. We also analyze the inherent non-stationarity of this multi-agent automatic curriculum teaching problem, and provide a corresponding regret bound. Empirical results show that our method improves scalability, sample efficiency, and generalization in multiple MARL environments. The source code and the video can be found at `https://sites.google.com/view/marl-spc/`.

## 1 INTRODUCTION

Multi-agent reinforcement learning (MARL) has long been a go-to tool in complex robotic and strategic domains (RoboCup, 2019; OpenAI, 2019). However, learning effective policies with sparse reward from scratch for large-scale multi-agent systems remains challenging. One of the challenges is that the joint observation-action space grows exponentially with varying numbers of agents. Meanwhile, the sparse reward signal requires a large number of training trajectories. Hence, applying existing MARL algorithms directly to complex environments with a large number of agents is not effective. In fact, they may produce agents that do not collaborate with each other even when it is of significant benefit (Zhang et al., 2021; Yang & Wang, 2020).

There are several lines of work related to the large-scale MARL problem with sparse reward, including: reward shaping (Hu et al., 2020), curriculum learning (Chen et al., 2021), and learning from demonstrations (Huang et al., 2021). Among these approaches, the curriculum learning paradigm, in which the difficulty of experienced tasks and the population of training agents progressively grow, shows particular promise. In *automatic* curriculum learning (ACL), a teacher (curriculum generator) learns to adjust the complexity and sequencing of tasks faced by a student (curriculum learner). Several works have even proposed *multi-agent* ACL algorithms, based on approximate or heuristic approaches to teaching, such as DyMA-CL (Wang et al., 2020c), EPC (Long et al., 2020), and VACL (Chen et al., 2021). However, DyMA-CL and EPC rely on a framework of an off-policy student with replay buffer, and ignore the forgetting problem that arises when the agent population size grows. ACL relies on the strong assumption that the value of the learned policy does not change when agents switch to a different task. Moreover, the teacher in these approaches is still facing an unmitigated non-stationarity problem due to the ever-changing student strategies. In addition, if we somewhat expand the ACL paradigm and presume that the teacher may have another purpose for the sequence

of tasks performed by the student, another class of larger-scale MARL solutions should be mentioned. Namely, hierarchical MARL, which learns temporal abstraction with more dense rewards, including: skill discovery (Yang et al., 2019), option as response (Vezhnevets et al., 2019), role-based MARL (Wang et al., 2020b), and two levels of abstraction (Pang et al., 2019). Alas, hierarchical MARL mostly focuses on one specific task with a fixed number of agents and does not consider the transfer ability of learned complementary skills. In this paper, we informally give our answer to a question:

*Can an elaborate combination of ACL and hierarchical principles learn **complex** cooperation with sparse reward in **MARL**?*

Specifically, we present a novel automatic curriculum learning algorithm, Skilled Population Curriculum (SPC), which learns cooperative behaviors from scratch. The core idea of SPC is to encourage the student to learn skills from tasks with different numbers of agents. Motivation from the real world is team sports, where players often train their skills by gradually increasing the difficulty of tasks and the number of coordinating players. In particular, we implement SPC with three key components with the teacher-student framework. First, to solve the final complex cooperative tasks, we model the teacher as a contextual bandit, where we utilize an RNN-based (Hochreiter & Schmidhuber, 1997) imitation model to represent student policies and use this to generate the bandit's context. Second, to handle the varying number of agents across these tasks, motivated by the transformer (Vaswani et al., 2017), which can process sentences of varying lengths, we implement population-invariant communication by treating each agent's message as a word. Thus, a self-attention communication channel is used to support an arbitrary number of agents sharing their messages. Third, to learn transferable skills in the sparse reward setting, we utilize the skill framework in the student. Agents communicate on the high level about a set of shared low-level policies. Empirical results show that our method achieves state-of-the-art performance in several tasks in the multi-particle environment (MPE) (Lowe et al., 2017) and the challenging 5vs5 competition in Google Research Football (GRF) (Kurach et al., 2019).

## 2 PRELIMINARIES

**Dec-POMDP.** A cooperative MARL problem can be formulated as a *decentralized partially observable Markov decision process* (Dec-POMDP) (Bernstein et al., 2002), which is described as a tuple $\langle n, \boldsymbol{S}, \boldsymbol{A}, P, R, \boldsymbol{O}, \boldsymbol{\Omega}, \gamma \rangle$, where $n$ represents the number of agents. $\boldsymbol{S}$ represents the space of global states. $\boldsymbol{A} = \{A_i\}_{i=1,\cdots,n}$ denotes the space of actions of all agents. $\boldsymbol{O} = \{O_i\}_{i=1,\cdots,n}$ denotes the space of observations of all agents. $P : \boldsymbol{S} \times \boldsymbol{A} \to \boldsymbol{S}$ denotes the state transition probability function. All agents share the same reward as a function of the states and actions of the agents $R : \boldsymbol{S} \times \boldsymbol{A} \to \mathbb{R}$. Each agent $i$ receives a private observation $o_i \in O_i$ according to the observation function $\boldsymbol{\Omega}(s, i) : \boldsymbol{S} \to O_i$. $\gamma \in [0, 1]$ denotes the discount factor.

**Multi-armed Bandit.** Multi-armed bandits (MABs) are a simple but very powerful framework that repeatedly makes decisions under uncertainty. In an MAB, a learner performs a sequence of actions. After every action, the learner immediately observes the reward corresponding to its action. Given a set of $K$ actions and a time horizon $T$, the objective is to maximize its total reward over $T$ rounds. The regret is used to measure the gap between the cumulative reward of an MAB algorithm and the best-arm benchmark. An representative work is the Exp3 algorithm (Auer et al., 2002), which is proposed to increase the probability of pulling good arms and achieves a regret of $O(\sqrt{KT\log(K)})$ under a time-varying reward distribution. Another related work is the contextual bandit problem (Hazan & Megiddo, 2007), where the learner makes decisions based on some prior information as the context.

## 3 SKILLED POPULATION CURRICULUM

In this section, we first provide a formal definition of the curriculum-enhanced Dec-POMDP framework, which formulates the MARL with curriculum problem under the Dec-POMDP framework. We then present our multi-agent automatic curriculum learning algorithm named Skilled Population Curriculum (SPC) as shown in Fig. 1. In the following subsections, we first formulate the curriculum learning framework in 3.1, then propose a contextual multi-armed bandit algorithm as the teacher to deal with the non-stationarity in 3.2, and last introduce the student with a skill and population-invariant communication framework to tackle the varying number of agents and the sparse reward problem in 3.3.

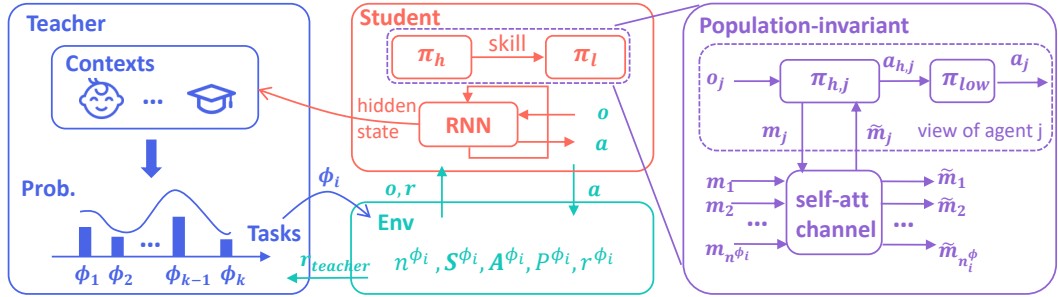

Figure 1: The overall framework of SPC. It consists of three parts: configurable environments, a teacher, and a student. Left. The teacher is modeled as a contextual multi-armed bandit. At each teacher timestep, the teacher chooses a training task from the distribution of bandit actions. Mid. The student is endowed with population-invariant communication and a skill framework, and trained with MARL algorithms on the training task. The student returns to the teacher not only the hidden state of RNN imitation model as contexts but also the average discounted cumulative rewards on the testing task. Right. The student learns hierarchical policies. The population-invariant communication is on the high level, and implemented with a self-attention communication channel to handle the messages from varying number of agents. The agents in the student share the same low-level policy.

## 3.1 PROBLEM FORMULATION

We consider environments from multi-agent automatic curriculum learning problems are equipped with parameterized task spaces and thus can be modeled as curriculum-enhanced Dec-POMDPs.

**Definition 3.1** (Curriculum-enhanced Dec-POMDP). A curriculum-enhanced Dec-POMDP is defined by a tuple $\langle \Phi, \mathcal{M} \rangle$, where $\Phi$ and $\mathcal{M}$ represent a task space and a Dec-POMDP, respectively. Given the task $\phi$, the Dec-POMDP $\mathcal{M}(\phi)$ is presented as $\{n^\phi, \boldsymbol{S}^\phi, \boldsymbol{A}^\phi, P^\phi, r^\phi, O^\phi, \boldsymbol{\Omega}^\phi, \gamma^\phi\}$. The superscript $\phi$ denotes that the Dec-POMDP elements are determined by the task $\phi$. Note that task $\phi$ can be a few parameters of the environment or task IDs in a finite task space. *In a curriculum-enhanced Dec-POMDP, the objective is to improve the student's performance on the target tasks through the sequence of training tasks given by the teacher.*.

Let $\tau$ denote a trajectory whose unconditional distribution $\mathrm{Pr}_\mu^{\pi,\phi}(\tau)$ under a policy $\pi$ and a task $\phi$ with initial state distribution $\mu(s_0)$ is $\mathrm{Pr}_\mu^{\pi,\phi}(\tau) = \mu(s_0) \sum_{t=0}^\infty \pi(a_t \mid s_t) P^\phi(s_{t+1} \mid s_t, a_t)$. We use $p(\phi)$ to represent the distribution of target tasks and $q(\phi)$ to represent the distribution of training tasks at each task sampling step. We consider the joint agents' policies $\pi_\theta(a|s)$ and $q_\psi(\phi)$ parameterized by $\theta$ and $\psi$, respectively. The overall objective to maximize in a curriculum-enhanced Dec-POMDP is:

$$J(\theta, \psi) = \mathbb{E}_{\phi \sim p(\phi), \tau \sim \mathrm{Pr}_\mu^\pi} \left[ R^\phi(\tau) \right] = \mathbb{E}_{\phi \sim q_\psi(\phi)} \left[ \frac{p(\phi)}{q_\psi(\phi)} V(\phi, \pi_\theta) \right] \tag{1}$$

where $R^\phi(\tau) = \sum_t \gamma^t r^\phi(s_t, a_t; s_0)$ and $V(\phi, \pi_\theta)$ represent the value function of $\pi_\theta$ in Dec-POMDP $\mathcal{M}(\phi)$. However, when optimizing $q_\psi(\phi)$, we cannot get the partial derivative $\nabla_\psi J(\theta, \psi) = \nabla_\psi \sum_\tau \frac{1}{q_\psi(\phi)} R^\phi(\tau) \mathrm{Pr}_\mu^{\pi,\phi}(\tau)$[1] since the reward function and the transition probability function w.r.t number of agents are non-parametric, non-differentiable, and discontinuous in most MARL scenarios.

Thus, we use the non-differentiable method, i.e., multi-armed bandit algorithms, to optimize $q_\psi(\phi)$, and use an RL algorithm (the student) in alternating periods to optimize $\pi_\theta(a|s)$. However, there are three key challenges in solving this problem: (1) There is a lack of a general student framework to deal with the varying number of agents across tasks and the sparse reward problem. (2) The teacher is facing a non-stationarity problem due to the ever-changing student's strategies. (3) The student will forget the old tasks and need to re-learning them. Some tasks can be the prerequisites of other tasks, while some can be inter-independent and parallel.

---

[1] $p(\phi)$ is not in the partial derivative since it is a fixed distribution.

## 3.2 TEACHER MODELED AS A NON-STATIONARY CONTEXTUAL BANDIT

As mentioned in the previous subsection, the teacher is facing a non-stationarity problem due to the ever-changing student's strategies, since the student learns across different tasks in process. That is, in different stages of student learning, the teacher will observe different student's performances when giving the same task to the student, thus leading to a time-varying reward distribution of the teacher. In addition, the student might forget the learned policy during the learning process. To avoid this problem, the teacher should offer some trained tasks to the student. It can be seen as the exploitation and exploration problem of the teacher. The teacher is encouraged to give the training tasks that benefit the student's performance on the target tasks; however, there is still a need for sufficient exploration on various training tasks that may not directly facilitate the student's learning.

Fortunately, we notice that the non-stationarity stems from the student, which can be mitigated with a contextual bandit which embeds the student policy into the context. As shown in Fig. 1 Left, the teacher takes the student's policy representation as the context and chooses a task from the distribution of training tasks. Specifically, we extend the Exp3 algorithm (Auer et al., 2002) with context by two-step online clustering (see Alg.1(Zhang et al., 1996)). The context $x$ is the student's policy representation, the teacher's action is a certain task $\phi$, and the teacher's reward is the return of the student in the target tasks. The teacher's algorithm is listed below. In steps 1-4, the teacher samples a task for the student's training. In steps 6-7, the teacher would update the parameters based on the evaluation reward of the student. So, we need to learn a good representation for the student's policy as the context.

---

**Algorithm 1** Teacher Sampling and Training

---

**Input:** Context $x$, the number of Clusters $N_c$, $N_c$ instances of Exp3 with task distribution $w(\phi_k, c)$ for $k = 1, \ldots, K$ and for $c = 1, \ldots, N_c$, learning rate $\alpha$, a buffer maintaining the historical contexts
**Output:** $\mathcal{M}(\phi) = \left\{ n^\phi, \boldsymbol{S}^\phi, \boldsymbol{A}^\phi, P^\phi, r^\phi, O^\phi, \boldsymbol{\Omega}^\phi, \gamma^\phi \right\}$, the teacher bandit parameters
**Sampling**
1. Get the the context $x$, and save it to the buffer
2. Run the online cluster algorithm and get the index of the cluster center $c(x)$
3. Let the active Exp3 instance be the instance with index $c(x)$
4. Set the probability $p(\phi_k, c(x)) = \frac{(1-\alpha)w(\phi_k, c(x))}{\sum_{j=1}^{K} w(\phi_k, c(x))} + \frac{\alpha}{K}$ for each task $\phi_k$
5. Sample a new task according to the distribution of $p_{\phi_k, c}$
**Training**
6. Get the return (discounted cumulative rewards) from student testing $r$
7. Update the active Exp3 instance by setting $w(\phi_k, c(x)) = w(\phi_k, c(x))e^{\alpha r/K}$

---

### 3.2.1 CONTEXT REPRESENTATION

A straightforward representation is to directly use the student parameters $\theta$ as the context. However, the number of parameters is too large to be used as the input of neural network if we change the student's architecture. Thus, we turn to an alternative method.

A principle to learn a good representation of a policy is *predictive representation*, that is, the representation should be accurate to predict policy actions given states. According to the principle, we utilize an imitation function through supervised learning. Supervised learning does not require direct access to reward signals, making it an attractive approach for reward-agnostic representation learning. Intuitively, the imitation function attempts to mimic low-level policy based on historical behaviors. In practice, we use an RNN-based imitation function $f_{im} : \mathcal{S} \times \mathcal{A} \to [0, 1]$. Since recurrent neural networks are theoretically Turing complete (Hyötyniemi, 1996), its internal states can be used as the representation of the student's policy. Regarding the training of this imitation function, we use the negative cross entropy objective $\mathbb{E}[\log f_{im}(s, a)]$.

### 3.2.2 REGRET ANALYSIS

In this subsection, we show the regret bound of the proposed teacher algorithm $\mathbb{E}[R(T)] = O\left(T^{2/3}(LK \log T)^{1/3}\right)$, where $T$ is the number of total rounds, $L$ is the Lipschitz constant, and $K$ is the number of arms (the number of the teacher's actions). The regret analysis is used to justify the

usage of the bandit algorithm in the non-stationary setting. Since the teacher's reward is the return of the student in the target tasks, the regret bound shows the optimality of the proposed method.

First, we introduce the Lipschitz assumption about the generalization ability of the task space.

**Assumption 3.2** (Lipschitz continuity w.r.t the context). Without loss of generality, the contexts are mapped into the $[0, 1]$ interval, so that the expected rewards for the teacher are Lipschitz with respect to the context.

$$|r(\phi \mid x) - r(\phi \mid x')| \leq L \cdot |x - x'|$$

for any arm $\phi \in \Phi$ and any pair of contexts $x, x' \in \mathcal{X}$ (2)

where $L$ is the Lipschitz constant, and $\mathcal{X}$ is the context space.

This assumption suggests that, for any policy that is trained on a set of tasks, the rate of performance change is not faster than the rate of policy change. It is a realistic assumption since we cannot expect the student to achieve a dramatic improvement on a given task when the student is represented by a new context via a few training steps.

Then, we borrow a contextual bandit algorithm for a small number of contexts (Auer et al., 2002) (see Appendix Alg. 2) and the lemma 3.3, as a stepping stone for the proof of Theorem 3.4.

**Lemma 3.3.** *Alg. 2 has regret* $\mathbb{E}[R(T)] = \mathcal{O}(\sqrt{TK|\mathcal{X}| \log K})$.

Lemma 3.3 introduces a square root dependence on $|\mathcal{X}|$ if running a separate copy of Exp3 for each context (Auer et al., 2002). It motivates us to handle large context space by discretization.

**Theorem 3.4.** *Consider the Lipschitz contextual bandit problem with contexts in* $[0, 1]$. *The Alg. 1 yields regret* $\mathbb{E}[R(T)] = O\left(T^{2/3}(LK \ln T)^{1/3}\right)$.

*Proof.* See Appendix B for the proof. □

In practice, the contextual space is high-dimensional instead of in $[0, 1]$, and a uniform mesh is used to discretize the context space in the proof. Since we cannot have such a uniform mesh, without loss of generality, we utilize the Balanced Iterative Reducing and Clustering using Hierarchies (BIRCH) online clustering algorithm (Zhang et al., 1996) to generate and discretize the context space. BIRCH summarizes large datasets into a smaller tree with clustered leaf nodes. The clustering is efficient and easy to update for newly added data. Thus, it is suitable for clustering the RNN-based policy representation. At the end of the training, the cluster can be seen as an approximation of the uniform mesh.

### 3.3 STUDENT WITH POPULATION-INVARIANT SKILLS

Here, to tackle the varying number of agents and the sparse reward problem, we propose a population-invariant skill framework where agents can communicate via a self-attention channel. In this framework, agents can learn the skills that can be transferred between different tasks. The student module is MARL algorithm-agnostic. It is orthogonal to any state-of-the-art MARL algorithm. Although there are few researches (Iqbal et al., 2021; Hu et al., 2021) trying to address the varying number of agents, they heavily rely on the prior knowledge of the environments.

**Population-Invariant Teamwork Communication.** Instead of learning independent policies for agents in the student, we introduce communication to enable the population-invariant property and learn tactics among agents. Motivated by the fact that the transformer (Vaswani et al., 2017) in natural language processing can handle varying lengths of sentences, we use the self-attention mechanism in our communication. As shown in Fig. 1 Right, each agent $j$ receives an observation $o_j$. In each round of communication, each agent $j$ sends a message vector $m_j = f(o_j)$ to a self-attention channel, where $f$ is an observation encoder function.

The channel aggregates all messages and sends the new message vector $\tilde{m}_j$ through the self-attention mechanism. Concretely, given the input of the channel $\mathbf{M} = [m_1, m_2, \cdots, m_n] \in R^{n \times d_m}$ and the trainable weight of the channel $\mathbf{W}_Q, \mathbf{W}_K, \mathbf{W}_V \in R^{d_m \times d_m}$, we can obtain three different representations $\mathbf{Q} = \mathbf{M}\mathbf{W}_Q, \mathbf{K} = \mathbf{M}\mathbf{W}_K, \mathbf{V} = \mathbf{M}\mathbf{W}_V$. Then, the output messages are

$$\tilde{\mathbf{M}} = \text{Attention}(\mathbf{Q}, \mathbf{K}, \mathbf{V}) = \text{softmax}\left(\frac{\mathbf{Q}\mathbf{K}^T}{\sqrt{d_m}}\right)\mathbf{V} = [\tilde{m}_1, \tilde{m}_2, \cdots, \tilde{m}_n] \quad (3)$$

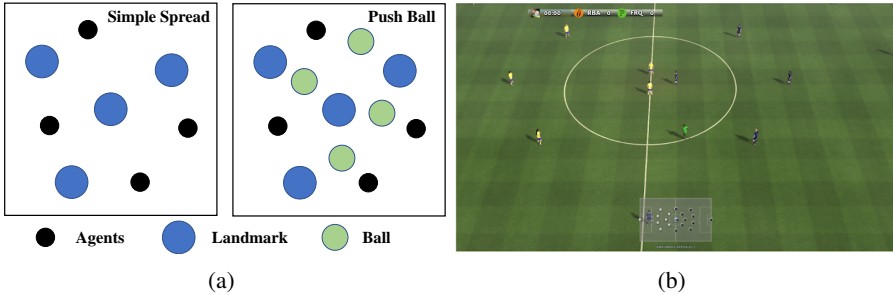

Figure 2: The environments. (a): Multi-particle Environment. (b): Google Reaserach Football

where $d_m$ is the dimension of the messages. Since the dimensions of the trainable weight are irrelevant to the number of agents, the student can take advantage of the population-invariant property to learn tactics.

**Transferable Hierarchical Skills.** As shown in the dotted box in Fig. 1 Right, after receiving the new messages $\tilde{m}_j$ from the channel, each agent takes the high-level action (skill) $a_{h,j} = \pi_{h,j}(o_j, \tilde{m}_j)$ to execute the low-level policy $a_j = \pi_{low}(o_j, a_{h,j})$. In this work, we generalize the high-level action (skill) $a_{h,j}$ to a continuous embedding space, so that the skill can be either a latent continuous vector as in DIAYN (Eysenbach et al., 2018), or a categorical distribution for sampling discrete options (Bacon et al., 2017).

**Implementation.** We implement the high- and low-level policies in the student with Proximal Policy Optimization (PPO) (Schulman et al., 2017). Following the common practice proposed in (Fu et al., 2022), the high-level policy for each agent is learned independently, whereas the low-level policies share parameters, since the most basic action pattern should be the same within different agents. The low-level agent is rewarded by the environment. The high-level policy takes actions given a fixed interval during training. Within this interval, a cumulative low-level reward is used as a high-level reward. When the categorical distribution is used to enable a discrete skill, we would sample an "option" from the categorical distribution and feed the corresponding one-hot embedding to the low-level policy.

## 4 FURTHER RELATED WORK

**Automatic Curriculum Learning in MARL.** Curriculum learning is a training strategy inspired by the human learning process, mimicking how humans learn new concepts in an orderly manner, usually based on the difficulty level of the problems (Portelas et al., 2020b). The selection of tasks is formulated as a Curriculum Markov Decision Process (CMDP) (Narvekar & Stone, 2018). Automatic Curriculum Learning mechanisms aim to learn a task selection function based on information about past interactions, such as ADR (Akkaya et al., 2019; Mehta et al., 2020), ALP-GMM (Portelas et al., 2020a), SPCL (Jiang et al., 2015), GoalGAN (Florensa et al., 2018), PLR (Jiang et al., 2021b;a), SPDL (Klink et al., 2020), CURROT (Klink et al., 2022), and graph-curriculum (Svetlik et al., 2017). Inspired by the mechanism of biodiversity in nature, a series of MARL curriculum learning frameworks have recently been proposed with remarkable empirical success. These include open-ended evolution (Banzhaf et al., 2016; Lehman et al., 2008; Standish, 2003), population-based training (Jaderberg et al., 2019; Liu et al., 2019), meta-learning (Gupta et al., 2021; Portelas et al., 2020c) and training with emergent curriculum (Baker et al., 2019; Leibo et al., 2019; Portelas et al., 2020b). In general, these frameworks can be unified under the idea of an automatic curriculum that automatically generates an endless procession of better performing agents by exerting selection pressure among many self-optimizing agents.

**Hierarchical MARL and Communication.** Hierarchical reinforcement learning (HRL) has been extensively studied to address the sparse reward problem and to facilitate transfer learning. Single-agent HRL focuses on learning the temporal decomposition of tasks, either by learning subgoals (Nachum et al., 2018b;a; Sukhbaatar et al., 2018; Nair & Finn, 2019; Wang et al., 2021) or by discovering reusable skills (Daniel et al., 2012; Gregor et al., 2016; Shankar & Gupta, 2020; Sharma et al., 2020). Recent works about hierarchical MARL have been discussed in the Introduction. In multi-agent settings, communication has demonstrated success in multi-agent cooperation (Foerster et al., 2016; Das et al., 2019; Sukhbaatar et al., 2016; Singh et al., 2018; Jiang & Lu, 2018; Kim et al., 2019; Wang et al., 2020a). However, existing approaches that extend HRL to multi-agent systems or

utilize communication are limited to a fixed number of agents and are hard to transfer with different number of agents.

**Google Research Football (Kurach et al., 2019).** Recent works attempt to tackle multi-agent scenarios in GRF by using a containerized learning framework (Wu et al., 2021), learning from demonstration (Huang et al., 2021), individuality (Jiang & Lu, 2021), and diversity (Li et al., 2021). However, they mainly focus on single-agent control, or train relatively easy academy tasks in GRF, or use offline expert data to train agents.

## 5 EXPERIMENTS

We consider several tasks in two environments, Simple-Spread and Push-Ball in the Multi-agent Particle-world Environment (MPE) (Lowe et al., 2017), and the challenging 5vs5 task of GRF (Kurach et al., 2019), to further demonstrate the performance of our approach.

We aim to answer the following three research questions. **Q1**: *Is curriculum learning needed in the complex large-scale MARL problem?* (Sec. 5.2) **Q2**: *Can our SPC outperform previous curriculum-based MARL methods? If so, which components in SPC contributes the most to performance gains?* (Sec. 5.3) **Q3**: *Can SPC learn a good curriculum for the student?* (Sec. 5.4)

### 5.1 ENVIRONMENTS, BASELINES AND METRIC

**Environments.** In the GRF 5vs5 scenario, we need to control 4 agents (except the goalkeeper) to compete with the opponent built-in AI. Each agent would observe a compact encoding, which consists of a 115-dimensional vector summarizing many aspects of the game, such as player coordinates, ball possession and direction, active player, and game mode. The action set available to an individual agent consists of 19 discrete actions such as idle, movement, passing, shooting, dribbling, or sliding. The GRF provides two types of reward: scoring and checkpoints, to encourage the agent to move the ball forward and have a successful shot.

In MPE, we investigate Simple-Spread and Push-Ball (see Fig. 2a). In Simple-Spread, there are $n$ agents that need to cover all $n$ landmarks. Agents are penalized for collisions and only receive a positive reward when all the landmarks are covered. In Push-Ball, there are $n$ agents, $n$ balls, and $n$ landmarks. The agents need to push the balls to cover every landmark. A success reward is given after all the landmarks have been covered.

**Baselines.** We evaluate the following approaches as baseline in Table 1:

We compare MARL algorithms to justify curriculum learning in the complex large-scale MARL problem. Also, we modify VACL by removing the centralized critic for a fair comparison of the MPE. Due to the difficulty of the GRF, we include a shooting reward to encourage the student to shoot.

Table 1: Baseline algorithms.

| Categories | Methods |
|---|---|
| MARL (**Q1**) | QMIX (Rashid et al., 2018) IPPO (de Witt et al., 2020) |
| Curriculum-based (**Q2**) | IPPO with uniform task sampling VACL (Chen et al., 2021) |
| Ablation Study (**Q3**) | SPC with uniform task sampling SPC without HRL and COM |

**Metric.** Even if we use the reward to optimize various algorithms, the mean episode reward in such environments cannot show the performance of the agents. Therefore, for GRF scenarios, we plot the win rate and the average goal difference, which is the number of goals scored by the MARL agents minus the number of goals scored by the other team.

The experiments are carried out on 30 nodes, one of which has a 128-core CPU and 4 A100 GPUs. Each experiment trial is repeated over 5 seeds and runs for 1-2 days.

### 5.2 THE NECESSITY OF CURRICULUM LEARNING

First, we describe experiments using MPE. In contrast to the fully observable setting and the centralized critic in VACL, we consider individual PPO in partially observable environments with default rewards. We randomly pick a starting state, and the episode ends after a fixed number of maximum steps. To be specific, the task space consists of $n$ agents, where $n \in \{2, 4, 8, 16\}$. We set the maximum allowed steps to 25. All evaluations are performed on the target task, where $n = 16$. IPPO is trained and evaluated directly on the target task. In Fig. 3, we can see that IPPO performs

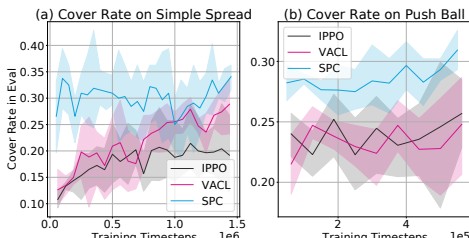 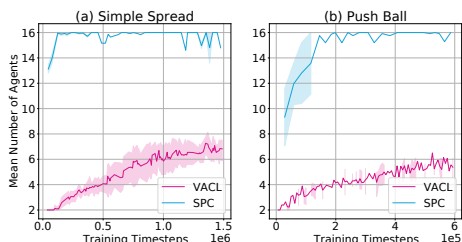

Figure 3: The evaluation performance of various methods on MPE.

Figure 4: The changes in the number of agents on MPE.

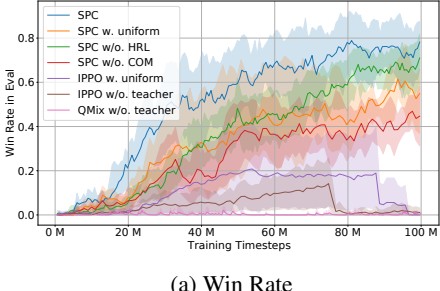 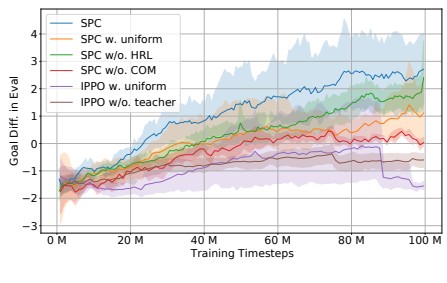

(a) Win Rate

(b) Goal Difference

Figure 5: The evaluation performance of various methods on 5vs5 football competition.

nearly VACL. SPC achieves a higher coverage rate than the baseline methods, but the improvement is not significant. Furthermore, we experimentally investigate the probability variation of different population sizes in Fig. 4. We observe that the curriculum afforded by SPC is approaching the target task. The results illustrate that in a simple environment where the student can directly learn to complete the task, there is no need to apply curriculum learning.

Then we show the performance comparison with the baselines in GRF. We also run CDS (Li et al., 2021) and CMARL (Wu et al., 2021), however, we did not include their performances, since the goal difference reported in CMARL (Wu et al., 2021) is relatively low compared to our method. In Fig. 5a , we can see that without the curriculum learning scheme, QMix and IPPO cannot perform well in the 5vs5 scenario. However, IPPO is slightly better than QMix in the scope of MARL algorithms in this scenario. In Fig. 5b, we omit the lines of QMix since the mean score is low, affecting the presentation of the figure. The reason could be that QMix is an off-policy MARL algorithm, which would rely heavily on the replay buffer. However, in such sparse reward scenarios, the replay buffer has much less efficient samples for QMix to learn. For example, the replay buffer would contain tons of zero-score samples, leading to a non-promising performance. Meanwhile, IPPO with a shared actor and critic, an on-policy algorithm, would utilize the samples more efficiently. Therefore, curriculum learning is a promising solution to the complex large-scale MARL problem.

During our experiments, we found that IPPO or shared parameter PPO can easily achieve good performance in most academic scenarios in GRF. However, 5vs5 is an obstacle for agents to handle more complex scenarios. Due to the limitation of computational resources, we tested SPC in the 11vs11 scenario. The result can be seen in the Appendix C.

## 5.3 PERFORMANCE AND ABLATION STUDY

In the experiments on MPE, In both environments, SPC performs better than VACL. Instead of training with continuous relaxation of the categorical distribution of population size in VACL, our bandit teacher achieves a higher success rate at test time, since the population size is a discrete variable in nature. Also, in Fig. 4, we observe that the curriculum provided by SPC is effective in exploring the task space as agents become increasingly competent.

In the experiments on GRF, we do not include VACL in our baselines in the GRF, since the implementation in the source code of VACL is heavily based on prior knowledge of specific scenarios, such as the threshold to divide the learning process. We can see that SPC has higher win rate and goal difference than IPPO with uniform task sampling in the 5vs5 football competition. The experiments on MPE and GRF show that when the teacher is rewarded by the student's performance,

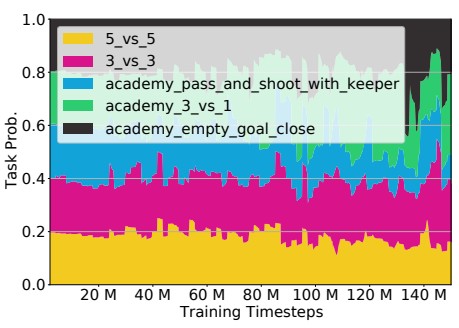

(a) The task distribution of SPC during training.

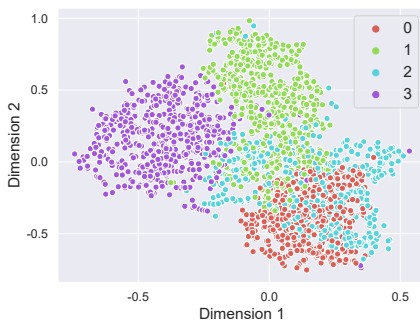

(b) The visualization of contexts

Figure 6: Visualization of Learned Curriculum.

the bandit-based teacher can exploit the student learning stage and give the suitable training tasks to the student.

For ablation study, we replace our contextual multi-armed bandit teacher with uniform task sampling and remove the hierarchical part in the student framework. As shown in Figs. 5a, 5b, we can clearly see that SPC can achieve a higher win rate and a greater score difference than SPC with uniform and SPC w/o. HRL. Also, SPC with uniform task sampling outperforms IPPO with uniform task sampling. The difference between these two methods is only the introduction of HRL. It shows the contribution of HRL in the 5vs5 football competition. When removing HRL and contextual multi-armed bandit, the performance degradation w.r.t. SPC are similar. It shows that HRL and the contextual multi-armed bandit seem to contribute equally. This can again justify the need for a curriculum learning scheme. However, we can see that SPC w. uniform has a larger variance in performance than SPC w/o. HRL. It means that uniform sampling might introduce more undesired tasks for student training.

## 5.4 VISUALIZATION OF LEARNED CURRICULUM

We visualize the distribution of task sampling of SPC during training based on a selected trial as shown in Fig. 6a. An interesting observation is that the task probability seems nearly uniform. We interpret this into an anti-forgetting mechanism. We can see that at the beginning of training, the task probability seems to be near-uniform, since the teacher should explore the task space and try to keep track of the student's learning status. During training, the probabilities vary over time steps. For example, at about 80-100 million timesteps, we can see a sudden drop in `academy_empty_goal_close` and `academy_3_vs_1_with_keeper`, since the student almost handles the skills learned in such scenarios. However, when training is continued, we can still observe that agents are trained on these tasks more frequently. We also visualize the distribution of contexts in Fig. 6b using t-SNE (Van der Maaten & Hinton, 2008). The contexts are collected and stored in a buffer. We divide the contexts into four classes according to the index. We can clearly see different parts in terms of different contexts about the final student policy representation.

## 6 DISCUSSION

**Conclusion.** We introduce a novel ACL algorithm, Skilled Population Curriculum (SPC). SPC addresses the scalability and sparse reward issue in current multi-agent system and learns complex behaviors from scratch. Specifically, to handle the varying number of agents, we incorporate a population-invariant multi-agent communication framework and exploit a hierarchical scheme for each agent to learn skills to deal with sparse rewards. Moreover, to mitigate the non-stationarity, we model the teacher as a contextual bandit, where the context is represented by the student's policy representation. Empirical results show that our method achieves state-of-the-art performance on several tasks in the multi-particle environment and the challenging 5vs5 competition in GRF.

**Limitations.** We acknowledge some limitations of our algorithm. First, since the objective of SPC is to solve difficulty tasks, SPC is over-designed when applied to some simple tasks. Second, the set of subtasks is still closely related to the environment. Lastly, it remains unclear how much the number of agents can affect the dynamics of the environment since we treat it as a black-box optimization.

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

# A ALGORITHM

---
**Algorithm 2** A contextual bandit algorithm for a small number of contexts
---
1: **Initialization:** For each context $x$, create an instance $\text{Exp3}_x$ of algorithm Exp3
2: **for** round **do**
3:     Invoke algorithm $\text{Exp3}_x$ with $x = x_t$
4:     Play the action chosen by $\text{Exp3}_x$
5:     Return reward $r_t$ to $\text{Exp3}_x$
6: **end for**
---

# B PROOF OF THEOREM 3.4

**Theorem 3.4.** *Consider the Lipschitz contextual bandit problem with contexts in $[0, 1]$. The Alg. 1 yields regret $\mathbb{E}[R(T)] = O\left(T^{2/3}(LK \ln T)^{1/3}\right)$.*

*Proof.* Let $S_m$ be the $\epsilon$-uniform mesh on $[0, 1]$, that is, the set of all points in $[0, 1]$ that are integer multiples of $\epsilon$. We take $\epsilon = 1/(d-1)$ where the integer $d$ is the number of points in $S_m$, which will be adjusted later in the analysis.

We apply Alg. 2 to the context space $S_m$. Let $f_{S_m}(x)$ be a mapping from context $x$ to the closest point in $S_m$:

$$f_{S_m}(x) = \min\left(\underset{x' \in S_m}{\text{argmin}} |x - x'|\right)$$

In each round $t$, we replace the context $x_t$ with $f_{S_m}(x_t)$ and call $\text{Exp3}_S$. The regret bound will have two components: the regret bound for $\text{Exp3}_S$ and (a suitable notion of) the discretization error. Formally, let us define the "discretized best response" $\pi_{S_m}^* : \mathcal{X} \to \Phi$: $\pi_{S_m}^*(x) = \pi^*\left(f_{S_m}(x)\right)$ for each context $x \in \mathcal{X}$.

We define the total reward of an algorithm Alg is $\text{Reward}(\text{Alg}) = \sum_{t=1}^{T} r_t$. Then the regret of $\text{Exp3}_S$ and the discretization error are defined as:

$$R_S(T) = \text{Reward}\left(\pi_S^*\right) - \text{Reward}\left(\text{Exp3}_S\right)$$
$$\text{DE}(S) = \text{Reward}\left(\pi^*\right) - \text{Reward}\left(\pi_S^*\right).$$

It follows that regret is the sum $R(T) = R_S(T) + \text{DE}(S)$. We have $\mathbb{E}\left[R_S(T)\right] = \mathcal{O}(\sqrt{TK \log K})$ from Lemma 3.3, so it remains to upper bound the discretization error and adjust the discretization step $\epsilon$.

For each round $t$ and the respective context $x = x_t$, $r\left(\pi_S^*(x) \mid f_S(x)\right) \geq r\left(\pi^*(x) \mid f_S(x)\right) \geq r\left(\pi^*(x) \mid x\right) - \epsilon L$. The first inequality is determined by the optimality of $\pi_S^*$ and the second is determined by Lipschitzness. Summing this up over all rounds $t$, we obtain $\mathbb{E}\left[\text{Reward}\left(\pi_S^*\right)\right] \geq \text{Reward}\left[\pi^*\right] - \epsilon LT$.

Thus, the regret is that

$$\mathbb{E}[R(T)] \leq \epsilon LT + O\left(\sqrt{\frac{1}{\epsilon}TK \log T}\right) = O\left(T^{2/3}(LK \log T)^{1/3}\right) \tag{4}$$

For the last inequality, we choose $\epsilon = \left(\frac{K \log T}{TL^2}\right)^{1/3}$. □

# C 11VS11 FULL GAME ON GRF

We further conduct experiments on the 11vs11 scenario of GRF. As shown in Fig. 7, SPC achieves about 50% win rate after training with 200 million timesteps.

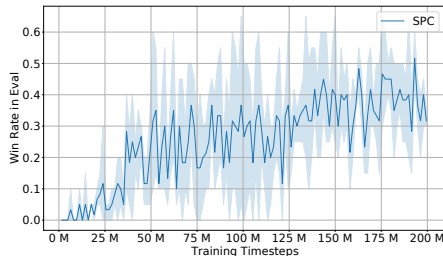

Figure 7: The performance of SPC on the 11v11 scenario.

# D    QUALITATIVELY ANALYSIS ABOUT LOW-LEVEL SKILLS

We show some statistics, for example, shooting, passing and running, in GRF in terms of different fixed low-level policies (defined by a set of fixed high-level actions). We evaluate the statistics by only fixing one agent's high-level action, maintaining other agents with trained SPC and testing 5 times. We show the results in Table 2.

|         | shooting per game | passing per game | running per game |
|---------|-------------------|------------------|------------------|
| skill 1 | 7.9 times         | 0.5 times        | 2254 time steps  |
| skill 2 | 2.3 times         | 26.4 times       | 2149 time steps  |
| skill 3 | 1.6 times         | 3.9 times        | 2875 time steps  |

Table 2: Statistics analysis about low-level skills.

# E    CORNER-5 ON GRF

To further illustrate the effectiveness of the SPC teacher module, we conduct experiments on the corner-5 scenario on GRF, where the target task is to control five of the eleven players to obtain a goal in the GRF `academy_corner` scenario. The experiments are designed to determine whether or not the contextual bandit in SPC outperforms alternative curriculum learning methods to schedule the number of agents in training. We compare SPC teacher against non-curriculum training (None), uniform task sampling (Uniform), ALP-GMM, and VACL. The training task space consists of $n$ agents, where $n \in \{1, 3, 5\}$. All teachers have the same base architecture without transformer architecture and hierarchical framework. We also investigate the effect of RNN-based contexts (see Contextual Bandit and Bandit). Fig. 8 shows the benefit of SPC contextual bandit over other ACL methods after training with one million timesteps.

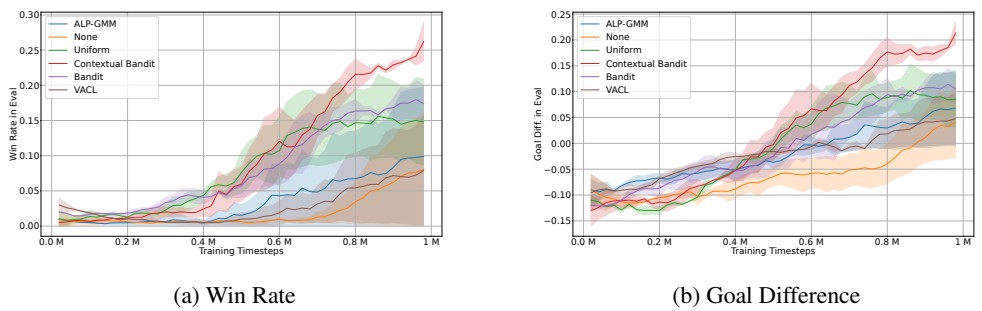

(a) Win Rate                    (b) Goal Difference

Figure 8: The evaluation performance of various teacher algorithms on the GRF corner-5 scenario.

Table 3: SPC hyper-parameters used in GRF.

| Name | Value |
| --- | --- |
| Discount rate | 0.99 |
| GAE parameter | 1.0 |
| KL coefficient | 0.2 |
| Rollout fragment length | 1000 |
| Training batch size | 100000 |
| SGD minibatch size | 10000 |
| # of SGD iterations | 60 |
| Learning rate | 1e-4 |
| Entropy coefficient | 0.0 |
| Clip parameter | 0.3 |
| Value function clip parameter | 10.0 |

Table 4: SPC hyper-parameters used in MPE.

| Name | Value |
| --- | --- |
| Discount rate | 0.99 |
| GAE parameter | 1.0 |
| KL coefficient | 0.5 |
| # of SGD iterations | 10 |
| Learning rate | 1e-4 |
| Entropy coefficient | 0.0 |
| Clip parameter | 0.3 |
| Value function clip parameter | 10.0 |

## F  IMPLEMENTATION DETAILS

Here we describe the SPC framework. We use the open-sourced Ray RLlib implementation of Proximal Policy Optimization (PPO), which scales out using multiple workers for experience collection. This allows us to use a large amount of rollouts from parallel workers during training to ameliorate high variance and aid exploration. We do multiple rollouts in parallel with distributed workers and use parameter sharing for each agent. The trainer broadcasts new weights to the workers after their synchronous sampling. We now turn our attention to environment-specific settings.

### F.1  GOOGLE RESEARCH FOOTBALL

We set five tasks for training the 5vs5 scenario. They are `academy_empty_goal_close`, `academy_pass_and_shoot_with_keeper`, `3_vs_3`, `academy_3_vs_1_with_keeper`, `5_vs_5`. In all scenarios, we do not control our team's goalkeeper.

In the `academy_empty_goal_close`, one agent need to move forward and shoot with an empty goal. In `academy_pass_and_shoot_with_keeper` and `3_vs_3`, two agents are controlled to play against a goalkeeper and 3 players respectively. In `academy_3_vs_1_with_keeper`, three agents are controlled to play against a center-back and a goalkeeper. In `5_vs_5`, 4 agents are controlled to play against 5 players. Without loss of generality, we initialize all player with fixed positions and roles as center midfielders.

We use both MLP and self-attention mechanism for the high-level policy, and use MLP for the low-level policy. For high-level policy, the input is first projected to an embedding using 2 hidden layers with 256 units each and ReLU activation, which is then fed into multi-head self-attention (8 heads, 64 units each). The output is then projected to the actions and values using another fully connected layer with 256 units. For low-level policy, we use MLP with 2 hidden layers with 256 units each, i.e., the default configuration of policy network in RLlib.

F.2    MPE

In this environment, agents must cooperate through physical actions to reach a set of landmarks. Agents observe the relative positions of other agents and landmarks, and are collectively rewarded based on the proximity of any agent to each landmark. In other words, the agents have to 'cover' all of the landmarks. Further, the agents occupy significant physical space and are penalized when colliding with each other. The agents need to infer the landmark to cover and move there while avoiding other agents.

The hyper-parameters of SPC in MPE are shown in Table 4. In MPE, hyper-parameters such as rollout fragment length, training batch size and SGD minibatch size are adjusted according to horizon of the scenarios so that policy are updated after episodes are done. We use the same network as in GRF, but with 128 units for all MLP hidden layers. Other omitted hyper-parameters follow the default configuration in RLlib PPO implementation.

