# OpenReview forum: "Towards Skilled Population Curriculum for MARL"
_ICLR.cc/2023/Conference — Submitted to ICLR 2023_

### Official Review · Reviewer_qKoe · 2022-10-20

**Confidence:** 4
**Correctness:** 3
**Technical Novelty And Significance:** 2
**Empirical Novelty And Significance:** 2
**Recommendation:** 6

**Clarity, Quality, Novelty And Reproducibility:**

- The novelty of the paper is satisfactory, as it is one of the few in the literature analyzing the use of curricula in MARL.
- Clarity of the paper could be significantly improved (see above)
- The source code of the method is included in the submission, which is good for reproducibility.

**Strength And Weaknesses:**

Strengths
-----------
- The paper considers the important problem of enabling the use of curricula in multi-agent reinforcement learning (MARL).
- The proposed solution is sound and backed up by satisfactory intuitive motivations.
- The empirical analysis is conducted on challenging RL experiments comparing with a satisfactory number of baselines. I think it would be a nice addition to have comparisons with more than the single QMIX method for value function factorization (e.g., QTRAN [1], QPLEX [2]), although I do not expect great improvements.

Weaknesses
--------------
- The paper is not very well written, and the structure is quite chaotic. Especially Section 3 is split in different subsections, not well connected, and the whole flow is not fluent.
- The regret bound could be an interesting theoretical results, but it is not complemented by any remark or empirical study, which makes the analysis a bit pointless.
- The method seems not easy to implement and relies on complex self-supervised mechanisms which may be not easy to use in generic problems. To this end, at least a hint of a computational and time complexity analysis of the proposed method is a needed addition.
- The ability to transfer seems an important feature of the proposed method, but experiments do not provide clear evidence of effective skills transfer.
- The paper does not cite important works in automatic curricula generation such as [3], [4], [5].

[1] Son, Kyunghwan, et al. "Qtran: Learning to factorize with transformation for cooperative multi-agent reinforcement learning." International conference on machine learning. PMLR, 2019.

[2] Wang, Jianhao, et al. "Qplex: Duplex dueling multi-agent q-learning." arXiv preprint arXiv:2008.01062 (2020).

[3] Klink, Pascal, et al. "Self-paced deep reinforcement learning." Advances in Neural Information Processing Systems 33 (2020): 9216-9227.

[4] Klink, Pascal, et al. "Curriculum Reinforcement Learning via Constrained Optimal Transport." International Conference on Machine Learning. PMLR, 2022.

[5] Svetlik, Maxwell, et al. "Automatic curriculum graph generation for reinforcement learning agents." Proceedings of the AAAI Conference on Artificial Intelligence. Vol. 31. No. 1. 2017.

**Summary Of The Paper:**

This paper introduces a feasible way to perform curriculum learning in multi-agent reinforcement learning (MARL). It describes well why curriculum learning cannot be applied to MARL seamlessly and proposes sound solutions to these problems. The experimental evaluation is conducted in a satisfactory way, although the results are not outstanding. There are additional theoretical results on the regret analysis of the proposed approach.

**Summary Of The Review:**

This paper is a possibly good contribution to MARL, but has currently some limitations that I described above and that I would like to see addressed in the revision. I am willing to increase my score in case of an improved version of this paper.

Post-rebuttal feedback
------------------------------
I thank the authors for the improved version of their submission. Most of my concerns have been addressed and I am happy to increase my score, as previously promised.

---

> ### Author Response · Authors · 2022-11-15
> **Response to Reviewer qKoe**
>
> Thank you for carefully reviewing our paper! We greatly appreciate your feedback. Please see below our responses to your comments.
>
> --------------
>
> **Q1**: Clarity about Section 3.
>
> **A1**: In Sec 3, we first introduce the problem in Sec 3.1, then show the teacher and student framework in Sec. 3.2 and 3.3 respectively. We add conjunction in the Sec 3 in the revised version.
>
> --------------
>
> **Q2**: Regret bound.
>
> **A2**: The regret analysis is used to justify the usage of the bandit algorithm in the non-stationary setting. Even though there is no direct empirical study, it allows us to use  online clustering algorithms to learn the ever-changing contexts in the non-stationary setting.
>
> --------------
>
> **Q3**: Reproducibility and computational and time complexity analysis.
>
> **A3**: We implemented SPC based on RLlib [1]. For generic problems, the introduction of the new environment takes more time, while the algorithm (SPC) can be directly used. The time complexity for running SPC in GRF is 100 million steps per day with a 128-core CPU and an A100 GPU. Other algorithms have similar time complexity in GRF. The bottleneck is in the rollouts of the environments instead of algorithms.
>
> --------------
>
> **Q4**: Transfer
>
> **A4**: The transferability lies in the student’s low-level policy. We include a video about the learned policy in other scenarios.
>
> --------------
>
> **Q5**: Related work.
>
> **A5**: We include and discuss these papers in the revised version.
>
> --------------
>
> **Q6**: Qplex adn Qtran
>
> **A6**: We refer to Fig.3 in [1] about the experiments of Qplex in GRF 5v5. The value-decomposition methods seemed not well-performed. We also mentioned other value-decomposition methods, e.g. CDS [2] and CMARL [1] in Sec 5.2.
>
> --------------
>
> [1] Containerized Distributed Value-Based Multi-Agent Reinforcement Learning. 2021.
>
> [2] Celebrating diversity in shared multi-agent reinforcement learning. 2021.

---

### Official Review · Reviewer_cbi7 · 2022-10-24

**Confidence:** 3
**Correctness:** 3
**Technical Novelty And Significance:** 3
**Empirical Novelty And Significance:** 3
**Recommendation:** 6

**Clarity, Quality, Novelty And Reproducibility:**

Clarity/Quality: The paper is overall written well. However, here are a couple suggested rephrasings:
- “by the teacher’s giving the sequence of training tasks” -> “through the sequence of training tasks given by the teacher”
- “It is diagonal to” -> “It is orthogonal to”

Other comments:
- Why is only a subset of methods evaluated in each environment? That is, why is QMix missing from MPE and VACL missing from GRF?
- It’s a good idea to report the performance of baselines even if they are low for completeness.
- For better clarity, I would recommend smoothing out the training curves in Figure 5 a bit.

Novelty: While the application of HRL to MARL settings is not new, this paper also uses contextual bandits to produce non-uniform task sampling distributions and a self-attention channel to combine agent messages, which are novel as far as I’m aware.

Reproducibility: The authors include code for their method and the baselines, which allow readers to reproduce the experiments in the paper.

**Strength And Weaknesses:**

Strengths:
- This paper tackles a particularly difficult problem, which is cooperative multi-agent RL from sparse rewards. While other methods struggle on the evaluated tasks, especially Google Research Football, SPC makes significant improvements in terms of performance.
- The visualizations and ablation study help readers understand the importance of each component of SPC.

Weaknesses:
- The experiments are only performed on Multi-agent Particle-world Environment and Google Research Football, which is relatively narrow compared to prior works that also study other environments like the StarCraft II benchmark.
- The method is quite complex. It’s shown, though, that the HRL structure and contextual bandit teacher are important. What about the self-attention channel?
- It would be useful to understand qualitatively what the different learned skills are doing.


**Summary Of The Paper:**

This paper proposes Skilled Population Curriculum (SPC), a framework for cooperative multi-agent reinforcement learning based on automatic curriculum learning. In particular, the student is equipped with a hierarchical skill set and designed to be population-invariant. The teacher is a contextual bandit that uses a representation of the student’s policy as its context. To implement population invariance, SPC uses self-attention to combine agent messages. In the experimental evaluation, SPC is compared to QMIX, IPPO, and VACL on Google Research Football and the Multi-agent Particle-world Environment, which they solve from (relatively) sparse rewards.

**Summary Of The Review:**

The experimental results are very strong, showing the improved performance achieved by SPC. There are also useful ablations and visualizations that illustrate how SPC achieves this performance. I am recommending to accept this paper, but believe it could be strengthened (see Weaknesses).

---

> ### Author Response · Authors · 2022-11-15
> **Response to Reviewer cbi7**
>
> Thank you for carefully reviewing our paper! We greatly appreciate your feedback. Please see below our responses to your comments.
>
> --------------
>
> **Q1**:  The StarCraft II benchmark.
>
> **A1**:  SMAC environment is a battle-based environment. The agents are encouraged to attack enemies one by one. In this case, the agents are supposed to have same behavior. SPC aims to learn complex cooperation with sparse reward in MARL. SOTA algorithms already can learn the hardest scenarios in SMAC without curriculum, while cannot learn in GRF 5v5. So we did not include SMAC.
>
> --------------
>
> **Q2**: Transformer/self-attention channel.
>
> **A2**: Transformer is used for communication. We remove the communication and show the results in Fig.5 in the revised version. We can see that the when communication is removed, the performance of SPC is not good as other version of SPC. It shows that in such complex games, the communication is more important for agents to cooperate.
>
> --------------
>
> **Q3**: Understand qualitatively what the different learned skills are doing.
>
> **A3**: Figure 6b showed the final clusters about the student policy representation. We add some statistics in GRF in terms of different fixed low-level policies in Appendix.
>
> --------------
>
> **Q4**: QMix missing from MPE and VACL missing from GRF.
>
> **A4**: QMix cannot have a comparable performance after training in MPE. VACL is heavily designed for MPE in their open-sourced code [1], e.g. VACL uses the position of each particle and the distance to compute the threshold in their value quantization (in their algorithm/autocurriculum.py).
>
> --------------
>
> **Q5**: Smoothing the training curves in Figure 5.
>
> **A5**: We revised Fig.5.
>
> ---------------
>
> [1] https://github.com/jiayu-ch15/Variational-Automatic-Curriculum-Learning.

---

> > ### Comment · Reviewer_cbi7 · 2022-11-18
> > **Response to authors**
> >
> > Thank you for the clarifications!

---

> > > ### Author Response · Authors · 2022-11-18
> > > **Re:  Response to authors**
> > >
> > > Dear Reviewer cbi7,
> > >
> > > We appreciate your suggestion to help us to improve this work. And if you have any further questions or comments, please post them and we will be happy to have further discussions.
> > >
> > > Paper 1827 authors

---

### Official Review · Reviewer_WEWH · 2022-10-24

**Confidence:** 4
**Correctness:** 2
**Technical Novelty And Significance:** 2
**Empirical Novelty And Significance:** 2
**Recommendation:** 6

**Clarity, Quality, Novelty And Reproducibility:**

**Clarity:** SPC has a relatively complex framework, so clarity needs improvement.
**Quality & Novelty:** SPC combines multiple existing frameworks, so novelty and contributions could be limited.
**Reproducibility:** The source code is provided in the supplementary material to reproduce the results.

**Strength And Weaknesses:**

**Strength:**
1. The paper develops the theoretical contribution: the regret analysis of the contextual bandit problem for solving ACL (Section 3.2.2).
2. Evaluations are performed using the complex domain of GRF.

**Weaknesses, Questions, and Comment:**
1. Overall, this paper combines multiple methods (e.g., transformer, skills, contextual bandits) for tackling too many challenges simultaneously, resulting in a relatively complicated framework.
2. While SPC ablation studies are provided, it is still challenging to understand the effectiveness of SPC. For example, SPC uses the transformer architecture, but baselines do not use the transformer. As such, it is unclear whether SPC performs better than baselines due to ACL and skill or due to the base architecture difference.
3. The paper's clarity can be improved. In particular, the paper states that multi-armed bandit algorithms are used due to the non-differentiable objective. However, there are other methods, including REINFORCE, and it will be helpful to justify why multi-armed bandit algorithms are selected over other methods.
4. Regarding hierarchical skills, the paper states that the categorical distribution is used to enable the option-style skill by passing the one-hot embedding to low-level policies. However, each option consists of a tuple of the initiation set, intra-option policy (low-level policy), and termination function. As such, it is unclear whether simply passing the one-hot embedding to the low-level policy enables the option.
5. Regarding the VACL baseline, the paper states that the centralized critic is removed for a fair comparison. However, my understanding is that the use of a transformer architecture can be effectively viewed as centralized training because my understanding is that gradient is passed across agents during training. If this view is correct, then it is unclear why the centralized critic is removed in VACL.
6. In Section 5.4, the paper states that Figure 6b shows the shift in student policy representation throughout the training, but this is unclear because Figure 6b does not include the time notion. Could you clarify this?
7. Minor: There is a typo: "In Fig. 6, we can see that without the curriculum learning scheme" -> "In Fig. 5, we can see that without the curriculum learning scheme"

**Summary Of The Paper:**

This paper proposes the SPC framework that combines automatic curriculum learning (ACL) and hierarchical MARL to learn cooperative behaviors in complex multi-agent domains. Specifically, this paper leverages a multi-armed bandit algorithm for ACL while addressing the following challenges: 1) the varying number of agents across tasks based on the transformer architecture, 2) non-stationarity based on the contextual bandit framework, and 3) reward sparsity based on the skills. Evaluations in MPE and GRF demonstrate the effectiveness of SPC.

**Summary Of The Review:**

I initially vote for 5 due to the weaknesses and concerns. I will make a final decision on the recommendation after the authors' response.

---

> ### Author Response · Authors · 2022-11-15
> **Response to Reviewer WEWH**
>
> Thank you for carefully reviewing our paper! We greatly appreciate your feedback. Please see below our responses to your comments.
>
> --------------
>
> **Q1**: Complicated framework.
>
> **A1**: Since we aim to solve the challenging cooperative MARL problems, we tested a few methods and found that only one method cannot be the silver bullet. As we mentioned in the Sec 3, there are three challenges: (1) a lack of a general student framework to deal with the varying number of agents; (2) a non-stationarity problem due to the ever-changing student's strategies; (3) the forgetting and relearning problem. These three challenges are related under the curriculum setting and not easy to be decoupled.
>
> --------------
>
> **Q2**: Transformer.
>
> **A2**: Transformer is used for communication. We remove the communication and show the results in Fig.5 in the revised version. We can see that the when communication is removed, the performance of SPC is not good as other version of SPC. It shows that in such complex games, the communication is more important for agents to cooperate.
>
> --------------
>
> **Q3**: Clarity about the usage of multi-armed bandits.
>
> **A3**: Many methods [1,2,3] in ACL can be used for solving the non-differentiable objective. Except that, we use bandits since the environment faced by the teacher doesn't have any state transitions and the actions are just a single choice from a fixed and finite set of choices.
>
> --------------
>
> **Q4**: Option-style skill.
>
> **A4**: Here we did not follow the option-critic strictly, instead, used the term “option-style skill” to show that the skill set is discrete. Since in option-critic, there are finite low-level policies. We did not include the initiation set and the termination function. In this way, one-hot embedding can represent $n$ skills where $n$ is the dimension of the one-hot embedding.
>
> --------------
>
> **Q5**: Centralized critic.
>
> **A5**: As stated in A2, a transformer is used for communication from the perspective of an actor. As shown in A2, we remove the communication and compare SPC and VACL without gradient-pass across agents.
>
> --------------
>
> **Q6**: Figure 6b.
>
> **A6**: Since we use the online clustering algorithm, thus the Figure 6b showed the final clusters. We modify the statement by saying: We can clearly see different parts in terms of different contexts about the final student policy representation.
>
> --------------
>
> **Q7**: Typos.
>
> **A7**: We fix these typos in the revised version.
>
> --------------
>
> [1] Emergent tool use from multi-agent autocurricula. 2019.
>
> [2] Autocurricula and the emergence of innovation from social interaction: A manifesto for multi-agent intelligence research. 2019.
>
> [3] Automatic curriculum learning for deep RL: A short survey. 2020.

---

> > ### Comment · Reviewer_WEWH · 2022-11-17
> > **Response to Rebuttal**
> >
> > I appreciate the authors for making the changes accordingly to my questions. The response mostly addresses my concerns. Overall, I agree with Reviewer cbi7 that SPC has positive evaluation results, but I also agree with Reviewer qKoe that the proposed method is complicated to follow and needs clarity improvement in writing. I update my score from 5 to 6, but my evaluation is still on the borderline.

---

> > > ### Author Response · Authors · 2022-11-18
> > > **Re: Response to Rebuttal**
> > >
> > > Dear Reviewer WEWH,
> > >
> > > We appreciate your constructive response to help us to improve this work. And thanks for pointing out the clarity issue.
> > >
> > > Based on your suggestion, we:
> > >
> > > - Modify the statement in the introduction about our methodology.
> > > - Add more conjunctions and transitions in Sec. 3.
> > > - We again check our typos and fix them.
> > >
> > > We thank you again and if you have any further questions or comments, please post them and we will be happy to have further discussions.
> > >
> > > Paper 1827 authors

---

### Official Review · Reviewer_HPXF · 2022-10-25

**Confidence:** 3
**Correctness:** 3
**Technical Novelty And Significance:** 4
**Empirical Novelty And Significance:** 3
**Recommendation:** 6

**Clarity, Quality, Novelty And Reproducibility:**

The proposed work is novel and the contribution is clearly highlighted. The paper although is a bit dense but is easy to read and the important relevant concepts are thoroughly explained. Some ablation studies as highlighted above could help bolster the claims of the paper.

**Strength And Weaknesses:**

The paper does an extensive study on the current issues in Automatic Curriculum Learning framework. The schematic diagram helps in better understanding the contribution and the proposed changes to the ACL framework. The MAB correction to train the teacher also helps in computing an approximate regret bound of its policy.

The paper misses out on some prior work that also use variable agents in a population as a means to learn diverse strategies [1, 2]. I am curious to know how the does the hierarchical setup affect in training diverse student strategies. Could a setup using meta-RL help in learning more diverse strategies rather than using bandit-based Exp3 algorithm?

Besides, some more ablation studies on analyzing the various student policies would also help in understanding the effect of the population invariant method?

[1] Gupta et al. 2021. Dynamic population-based meta-learning for multi-agent communication with natural language.

[2] Portelas et al. 2020. Meta Automatic Curriculum Learning.

**Summary Of The Paper:**

The paper proposes a learning framework that aims to address the change in the number of agents in an environment as a student agent is trained through a curriculum of tasks as proposed by a separate teacher agent. The teacher agent is modeled as a contextual bandit that aims to resolve the inherent non-stationarity in student policies. The results show that using the proposed changes, the agents are able to outperform previous work on competitive benchmarks.

**Summary Of The Review:**

The paper proposes novel changes to the popular ACL framework addressing key challenges such as non-stationarity in student policies and variable agents in the populations.

---

> ### Author Response · Authors · 2022-11-15
> **Response to Reviewer HPXF**
>
> Thank you for carefully reviewing our paper! We greatly appreciate your feedback. Please see below our responses to your comments.
>
> --------------
>
> **Q1**: Related work.
>
> **A1**: We include and discuss these two papers in the revised version.
>
> --------------
>
> **Q2**: The hierarchical setup effect in training diverse student strategies.
>
> **A2**: The hierarchical setup can help agent to learn transferable skills. Such skills are learned in different curriculums. So if we have diverse curriculums, then student can learn diverse strategies. Also, the mutual information normalization [1] can be used to encourage diversity.
>
> --------------
>
> **Q3**: Meta-RL helps in learning more diverse strategies.
>
> **A3**: Meta-RL mainly focuses on finetuning the hyperparameters [2] or adapting to new environments [3]. Since the meta-RL follows an inner-outer-loop manner, which is similar to the student-teacher framework, we believe meta-RL can be a solution and might lead to diverse strategies with a proper loss function as a future work.
>
> --------------
>
> **Q4**: more ablation studies.
>
> **A4**: We add new corner-kick experiments in football to show the effect of the population invariant method. Since the number of agents is different from the 5v5 scenario.
>
> ----------------
>
> [1] Diversity is All You Need: Learning Skills without a Reward Function. 2018.
>
> [2] Meta-Gradient Reinforcement Learning. 2018.
>
> [3] Model-Agnostic Meta-Learning for Fast Adaptation of Deep Networks. 2017

---

### Author Response · Authors · 2022-11-16
**Summary of the revised version**

We thank all reviewers for suggestions and constructive comments to help improve this work. If you have any further questions or comments, please post them and we will be happy to have further discussions.

Please see below our general responses and summary of the revised version.

Summary of the revised version (We use blue color to highlight the main changes):

- Modify Sec.3 for more clarity. (Q1 of Reviewer qKoe)
- Move related work about curriculum learning. (comments from Reviewer HPXF and qKoe)
- Add SPC without communication and smooth Fig 5. (Q2 of Reviewer WEWH and Q2, Q5 of Reviewer cbi7)
- Add corner experiments in Appendix. (Q4 of Reviewer HPXF and Q2 of Reviewer WEWH)
- Add qualitative statistics in GRF in terms of different fixed low-level policies in the Appendix. (Q3 of Reviewer cbi7)
- Other fixes about typos, grammar, and figures.

---

> ### Author Response · Authors · 2022-11-18
> **More revisions**
>
> Based on Reviewer WEWH's suggestion and his agreement with Reviewer qKoe, we:
>
> - Modify the statement in the introduction about our methodology.
> - Add more conjunctions and transitions in Sec. 3.
> - We again check our typos and fix them.
>
> We thanks all reviewers' suggestion and constructive comments to help us to improve this work. If you have any further questions or comments, please post them and we will be happy to have further discussions.
>
> Paper 1827 authors

---

### Decision · Program_Chairs · 2023-01-20

**Decision:**

Reject

**Justification For Why Not Higher Score:**

The paper was discussed among all the reviewers and inclined toward a rejection decision. The reviewers shared a common concern that the proposed method is not easy to implement as it involves several complex components (e.g., transformer, skills, contextual bandits). The paper does not give a clear justification for the complexity of the proposed method and whether all these components are necessary; moreover, these components in the proposed method may not be easy to apply in other multi-agent domains.

**Justification For Why Not Lower Score:**

N/A

**Metareview: Summary, Strengths And Weaknesses:**

The reviewers agreed that the paper considers an important setting of curriculum learning in multi-agent domains and acknowledged the novelty of the proposed curriculum method. However, the reviewers pointed out several weaknesses and shared a common concern that the proposed method is not easy to implement as it involves several complex components (e.g., transformer, skills, contextual bandits). The paper does not give a clear justification for the complexity of the proposed method and whether all these components are necessary; moreover, these components in the proposed method may not be easy to apply in other multi-agent domains. We want to thank the authors for their detailed responses; the paper was also discussed among all the reviewers, considering the responses and revision. Based on the raised concerns and follow-up discussions, unfortunately, the final decision is a rejection. Nevertheless, this is exciting and potentially impactful work, and we encourage the authors to incorporate the reviewers' feedback when preparing a future revision of the paper.